# Degradation Feature Extraction Method for Prognostics of an Extruder Screw Using Multi-Source Monitoring Data

**DOI:** 10.3390/s23020637

**Published:** 2023-01-05

**Authors:** Jun-Kyu Park, Howon Lee, Woojin Kim, Gyu-Man Kim, Dawn An

**Affiliations:** 1Renewable Energy Solution Group, Korea Electric Power Research Institute, Naju 58277, Republic of Korea; 2Advanced Mechatronics R&D Group, Korea Institute of Industrial Technology, Daegu 42994, Republic of Korea; 3School of Mechanical Engineering, Kyungpook National University, Daegu 41566, Republic of Korea

**Keywords:** degradation feature, data processing, prognostics, screw, extrusion system, real operational data, multi-source data, structural health monitoring

## Abstract

Laboratory-scale data on a component level are frequently used for prognostics because acquiring them is time and cost efficient. However, they do not reflect actual field conditions. As prognostics is for an in-service system, the developed prognostic methods must be validated using real operational data obtained from an actual system. Because obtaining real operational data is much more expensive than obtaining test-level data, studies employing field data are scarce. In this study, a prognostic method for screws was presented by employing multi-source real operational data obtained from a micro-extrusion system. The analysis of real operational data is more challenging than that of test-level data because the mutual effect of each component in the system is chaotically reflected in the former. This paper presents a degradation feature extraction method for interpreting complex signals for a real extrusion system based on the physical and mechanical properties of the system as well as operational data. The data were analyzed based on general physical properties and the inferred interpretation was verified using the data. The extracted feature exhibits valid degradation behavior and is used to predict the remaining useful life of the screw in a real extrusion system.

## 1. Introduction

Structural health monitoring (SHM) has been employed in various fields as a cost-effective maintenance strategy based on sensing systems, such as acoustic emission [1,2], piezoelectric [3,4], vibration [5,6], and multi-source sensors [7,8,9]. SHM data are utilized in diagnostics and prognostics. In diagnostics, they detect, isolate, and identify the damage and/or defect of a structure; in prognostics, they predict degradation behavior and remaining useful life (RUL) of an in-service system. In recent decades, prognostics have been studied in various engineering applications such as bearings [6], aircraft engines [7], batteries [8], and fuel cell stacks [9]. Recently, Huang et al. [6] developed novel methods for bearing prognostics but employed the open-source bearing datasets that are widely used for bearing prognostics. Studies on RUL prediction of aircraft engines by Liu et al. [7] used simulation datasets of turbofan engine degradation. This type of dataset is typically generated using simulation tools such as the commercial modular aero-propulsion system Simulation [10]. Zhang and Li [8] recently provided a detailed summary of lab-scale datasets used in the field of lithium-ion batteries. Marine et al. [9] conducted a prognostic study of a fuel cell (FC) stack using datasets obtained from aging tests, determining the effect of high-frequency current ripples on FC stack durability.

As mentioned above, most existing studies used data obtained under laboratory conditions; the datasets were obtained in an easy, fast, and cost-effective manner using an accelerated test or a well-organized test plan. Before applying the developed prognostic method to a real industrial field, it is essential to employ a simple dataset for pilot testing purposes. However, the prognostic method has rarely been validated with real operational data because obtaining field data is time-consuming and expensive. Run-to-failure data are required to validate the prognostic method; however, the life span of a real operating system is several years or even decades. Considering the differences in the process and operating conditions used to obtain laboratory and field data, the importance of employing real operational data cannot be sufficiently emphasized. 

In this study, real operational data were obtained from a micro-extrusion system for medical tube-grade catheters. To the best of our knowledge, in addition to using real operational data, no study has addressed the prognosis of the extrusion system or its components. It is important to maintain the quality of the product, particularly for medical catheters, as they can be applied to the human body. There have been long-standing efforts to maintain the extrusion product quality itself [11,12,13,14,15]; however, the maintenance strategy for the extrusion system is rather simplified, although its health condition influences the quality of products. Additional extrusion processes are performed as a maintenance strategy, which is periodically conducted under controlled conditions that restrict the process variables, raw materials, etc. [16]. Although this method is intuitive, it involves cumbersome operations that require additional time and money. Therefore, a prognostic method for an extrusion system is presented based on real operational data to monitor the health condition of the system without additional tasks.

The extrusion system consists of a variety of components, including the motor, screw, barrel, and puller, and its health condition is affected by various failure modes of the components. Among them, screw wear is the most common failure mode of the extrusion system. Further, it affects the mixing quality and temperature of the molten polymer, thereby affecting the quality of the products (catheters) [17,18]. The quality of products can be maintained under moderate wear of the screw by adjusting the process variables but cannot be maintained under severe wear of the screw. However, it is not easy to distinguish whether the level of screw wear is moderate or severe because it progresses gradually, and the quality of products also depends on other factors such as operator and environmental conditions. Inadequate control of the extrusion process due to wear can cause variations in product quality [19,20], which can lead to raw material wastage, increased energy consumption, and environmental pollution [21,22]. Therefore, this study aims to predict the RUL of the extrusion screw based on real operational data so that a timely replacement is performed before the screw under inadequate performance deteriorates the quality of the products.

The analysis of real operational data is more challenging than that of test-level data because it is obtained at the system level, wherein the mutual effect of each component on the system is reflected chaotically. Therefore, the main contribution of this study is the extraction of the degradation feature to monitor the wear of the screw used in a real extrusion system, which is based on the physical and mechanical properties of the system as well as the operational data. The extracted feature exhibits valid degradation behavior and is used to predict the RUL of the extrusion screw during its lifespan.

The remainder of this paper is organized as follows. In Section 2, the medical catheter extrusion system and the operating data are introduced. In Section 3, the process of extracting the degradation features of screw wear based on the physical interpretation of the extrusion system and experimental data is explained, and the features are applied to real operational data. In Section 4, the application of the extracted degradation features to predict the RUL of an extrusion screw is described. Finally, a brief conclusion is presented.

## 2. Extrusion System

An extrusion system is a continuous production system that uses polymer melts manufactured through frictional heat between a cylinder and a screw to produce a tube using a mold composed of a tip and die. An extrusion system is composed of several main components, such as a hopper, screw and barrel, tip and die, quenching system, vacuum water tank, measurement device, puller, cutter, and conveyor system. The actual micro-extrusion system (Davis-Standard Inc., Fulton, NY, USA) is shown in Figure 1. This provided the operational data used in this study. Detailed information on the extrusion process and operating data is provided in the following subsections.

### 2.1. General Extrusion Process and System Configuration

The extrusion system shown in Figure 1 is illustrated in Figure 2 to introduce the general extrusion process. After drying and dehumidifying, the polymer is injected into the barrel equipped with a screw through the hopper of the extruder and melted by frictional heat between the screw, polymer, and inner wall of the barrel. The initial shape of the tube is generated as the polymer passes through the tip and dies by adjusting the rotating speed of the screw, and the pressure of air injected into the lumen controls the ovality and shape of the tube. The lumen of the tube is stably hardened in a quenching part and vacuum tank filled with water, and the size of the tube is precisely controlled by adjusting the puller speed. The final tubes are produced on a conveyor system after the consecutive tubes are cut.

Many factors affect the quality of products during complex extrusion processes. The screw wear-related parts are illustrated in Figure 3 with the locations of the on-board sensors, which are used to monitor the condition of each component. The components are classified into two large groups: the extruder part, which includes the motor, screw in the barrel, tip, and die; and the puller part, which includes the puller itself. Four types of sensing signals (motor load, head pressure, and screw and puller speeds) were selected from the many on-board sensors attached to the actual extrusion system by considering their relationship to screw wear. In Figure 3, the numbers represent the sequence of the operational flow: 1. The screw speed setting (called S.Spd.Set in diagrams and calculations) is a control parameter that is adjusted to make the quality of the catheter close to the final product; 2. The motor operates by following the screw-speed setting. The motor load is denoted by M.Load; 3. The motor rotates the screw in the barrel, and the actual screw speed (S.Spd.Act) is monitored; 4. The rotating screw extrudes the polymer melt in the barrel to the tip and die, and the head pressure (H.Press) at this time is monitored. The coarsely shaped catheter is then moved to puller: 1*. the puller speed setting (P.Spd.Set), another control parameter, is precisely adjusted to make the catheter of high quality; 2*. the puller motor runs the puller following the puller speed setting, and the actual puller speed (P.Spd.Act) is measured. In this study, the operational data of the system were analyzed based on the operational mechanism shown in Figure 3.

### 2.2. Data Description

The extrusion system shown in Figure 1 has been operational since March 2017, and no component has been replaced (but minimum maintenance was conducted), except for screws. Two types of screws were used in the same extrusion system according to the polymer series. The screws used for the polyurethane/polyester and polyamide series are called barrier and (spiral) Maddock screws, respectively. The barrier screw has been in use since it was replaced in July 2020, and the Maddock screw has been in use since the start of the operation and is now severely worn. The operational data used in this study were obtained from July 2020 to March 2022. In summary, the barrier screw is likely to have mild wear, whereas the Maddock screw has reached its end-of-life (EOL). Therefore, the Maddock screw was considered the target component for prognostics in this study.

The extrusion system was operated for approximately eight hours per day on weekdays. The real operational data for all the measurements are shown in Figure 4a. In the figure, the bars represent the monitoring data for each day, but these raw data are not easy to interpret. This is the reason why feature extraction is required. The operational data for one day (green box) are shown in Figure 4b. In the figure, the extrusion process data can be categorized into two types, as shown in Figure 3: control (colored red) and measurement (colored blue) data. As shown in Figure 4b, all measurement values were changed in real time by following operator-controlled speed settings.

As shown in Figure 4b, the datasets were obtained in two different ways according to the operational conditions. First, a component-level experiment was conducted to monitor the effect of screw wear on extrusion data by minimizing other effects, such as motor degradation and processing variables. Next, real operational data were collected from an actual extrusion system that produces medical catheters over a long period. A more detailed explanation of the data is provided in the following subsection.

#### 2.2.1. Screw Experimental Data

Experimental screw wear data were obtained from three screws with different levels of wear under the same production conditions that were used to produce the same type of single-lumen catheters with an inner diameter of 2.0 mm and outer diameter of 2.5 mm using the same polymer (carbothane PC-3595A, Lubrizol, Cleveland, OH, USA) over two days. It should be noted that screw wear naturally progresses during actual use, as shown in Figure 5. In the figure, wear levels 1, 2, and 3 correspond to the intact, moderately worn, and completely worn screws, respectively. The experimental data, which are similar to the signals in Figure 4, are analyzed in Section 3.

#### 2.2.2. Real Operational Data

According to the production schedule, one of the barrier and Maddock screws were used in the same extrusion system shown in Figure 1, which means that the system and two screws have different life spans. Even though the operation of the extrusion system and the use of the Maddock screw started in March 2017, the time when the monitoring started (July 2020) was assumed to be the initial cycle. Consequently, the total operating times of the extrusion system and Maddock screw were approximately 4000 h and 1200 h, respectively. The cumulative usage time of the barrier screw was approximately 2800 h (which is the actual cumulative usage time rather than the assumed life of the system and Maddock screw), but it was not considered in this study. During the extrusion process, catheters with various specifications were produced using different polymer materials. These diversities made it difficult to extract the degradation features of screw wear from monitoring signals. In the following section, the robust wear feature was extracted regardless of the type of catheter produced and the material used.

## 3. Degradation Feature Analysis in the System

To predict the RUL of a screw, the degradation feature must first be extracted in the form of a monotonic increase or decrease. However, it is challenging to extract degradation features from raw data, particularly from real operational data, as shown in Figure 4a. Moreover, it may take several decades to obtain sufficient data from an actual system for prognostics, including for feature extraction. Therefore, data and physical information are complementarily used; raw data are analyzed based on physical and mechanical interpretation, and the inferred interpretation is, in turn, verified by the data.

### 3.1. Physical and Mechanical Properties of the Extrusion System

Six signals were recorded, including the control and measurement signals, as shown in Figure 3. Among the six signals, the setting and actual speeds were examined first, as shown in Figure 6. In the figure, the blue solid and red dashed lines represent the setting and actual speed, respectively. They were very close to each other during the normal extrusion process between the two green vertical lines at both screw and puller speeds. Thus, the health condition of the motor, monitored until the current time, had no valid effect on the following process (refer to Figure 3). In other words, the head pressure was not affected by motor degradation because the actual screw speed followed the set value well, regardless of the motor condition. 

However, the effect of motor degradation was reflected in the change in motor load, which usually increased as the motor degraded [23]. Screw wear was also related to the motor load, which decreased as the screw wore out because the screw speed increased owing to the reduced radius of the blade [24]. Note that the motor load decreased with increasing screw speed caused by wear but increased with an increase in the setting speed of the screw. Thus, the motor load reflected at least three aspects: the screw speed setting, screw wear, and motor degradation. This explains why the data analysis at the system level is complex and difficult.

The motor load was closely related to the mechanical aspect, whereas the head pressure and puller speed were more closely related to catheter quality. As screw wear progressed, head pressure typically decreased, and the screw and puller speeds were appropriately adjusted by the operator to maintain catheter quality. However, as shown in Figure 3 and Figure 4, screw speed had a greater effect on the overall system control than the quality of catheters. In general, to maintain catheter quality, puller speed should increase with a decrease in head pressure, which is based on the law of conservation of energy. Maintaining the same catheter quality required the same energy; however, energy loss occurred because of a decrease in head pressure as the screw wore out over time. Thus, the puller speed must be increased to compensate for lost energy.

To summarize the above two paragraphs, screw wear over time (t) can be expressed as follows:(1)$$d\left(t\right)=L\left(t\right)A\times \frac{P\left(t\right)}{V\left(t\right)}$$
where L, P, V, and d represent the motor load, head pressure, puller speed, and screw degradation level, respectively. The motor load in Equation (1) reflects the degradation of both the screw and motor and decreases or increases depending on the predominance of the two degradations. However, motor degradation was not considered in this study because it is negligible compared with the wear level of the Maddock screw. The head pressure and puller speed decreased and increased, respectively, as the screw wore out. In conclusion, each term on the right side of Equation (1) is an indicator that can be used to monitor screw deterioration. When screw deterioration is dominant, Equation (1) clearly shows a gradual decrease. The degradation feature of the screw wear in Equation (1) was verified and improved using experimental and real operating data in the following subsections.

### 3.2. Experimental Data Analysis

The most common phenomenon caused by screw wear is a decrease in the head pressure, which was verified through experiments, as shown in Figure 7a. The black and red signals were obtained from the screws of wear levels 1 and 3, respectively, as shown in Figure 5. This result was obtained experimentally by letting the polymer melt flow down at the tip and die with a fixed screw speed for both the intact and worn screws. As shown in Figure 7a, the head pressure of the intact screw (black) is approximately 2 MPa higher than that of the worn screw (red). Although it was obvious that the head pressure decreases with screw wear, the data from the real operating process for producing catheters, as shown in Figure 7b, did not support this conclusion. In the figure, the head pressures of the intact (black) and worn (red) screws are similar, and it is difficult to distinguish which dataset corresponds to the worn screw from the signals. This is because the operation settings, such as the screw and puller speeds, were adjusted to maintain product quality under different health conditions of the screws.

The experimental results using the three screws in Figure 5 are shown in Figure 8, where the black, blue, and red colors represent wear levels 1, 2, and 3, respectively. The operational data during normal extrusion are shown in Figure 8a, and the corresponding averages are shown in Figure 8b. The y-axes of the four results in Figure 8b were scaled to a 20% difference between the maximum and minimum values. In the figure, significant changes in the motor load and puller speed can be observed, whereas the head pressure did not show much change (note that the screw speed was fixed at 10.7 rpm).

As mentioned in Section 3.1, the motor load was affected by the screw speed setting and degradation of both the motor and screw. The screw speed was fixed during the experiment. In addition, the deterioration in motor performance cannot be reflected because the experiment was performed for two consecutive days. Therefore, we can conclude that the reduction in motor load in Figure 8b was caused only by screw degradation. However, Figure 8b does not show the expected result of the puller speed increasing continuously as the wear level increases. Despite the obvious screw wear, the puller speed increased at wear level 2, but decreased at wear level 3. To understand this phenomenon, an unexpected inflow of energy, such as fluctuation energy, was investigated by considering the severity of wear level 3.

The production quality of the catheters could not be maintained when a screw with a wear level of 3 was used. This was demonstrated by a flow-rate test and the quality of the produced catheter. The flow rate was measured for one minute at a fixed screw speed of 10.7 rpm according to the wear level listed in Table 1. The tests were performed three times each, and the average and standard deviation were listed with the rate of increase relative to wear level of 1. The flow rate was expected to increase as the physical space increased owing to an increase in screw wear; however, the flow rate at wear level 3 was lower than the flow rate at wear level 2. This is because the melted polymer adheres to the screw surface and prevents it from falling. The severity of wear level 3 led to a large variation in not only the standard deviation of the flow rate (a 652% increase over wear level 1), but also in the quality of the catheter, as shown in Figure 9. In the figure, the yellow circles are the catheters produced with the three wear levels of the screws, and the red circles that are the same for each figure represent the ideal shape and size of the catheter. As shown in the figure, the catheter produced using the screw of wear level 1 was very close to the red circle, and the catheter of wear level 2 was slightly distorted in the upper part (between two green lines). On the other hand, the catheter using the screw with wear level 3 showed a large discrepancy between the product and the right half of the red circle (between the two green lines). In conclusion, the wear level 3 screw was not suitable for producing a fair quality catheter. Therefore, the useful life of the screw should be considered between wear levels 2 and 3.

Before employing real operational data, an additional aspect should be considered. To date, experimental data have been obtained under conditions that produce the same types of catheters at a fixed screw speed using the same polymer. Because various types of catheters and polymers are used during actual operation, the degradation feature in Equation (1) must be modified to reflect these variabilities. The simplest method is to employ the screw speed setting corresponding to the overall system control as a correction factor as follows:(2)$$d_{M}\left(t\right)=\frac{1}{R\left(t\right)}A\times L\left(t\right)$$
(3)$$d_{E}\left(t\right)=\frac{1}{R\left(t\right)}A\times \frac{P\left(t\right)}{V\left(t\right)}$$
(4)$$d_{T}\left(t\right)=\frac{L\left(t\right)A\times P\left(t\right)}{R\left(t\right)A\times V\left(t\right)}$$
where $d_{M},\;d_{E},\;d_{T}$, and R represent the mechanical, energy, total degradations, and screw speed, respectively. The degradation feature in Equation (1) is divided into the mechanical aspect ($d_{M}$) in Equation (2) and the energy aspect ($d_{E}$) in Equation (3) because they show different behaviors at the very late stage of screw life. The total degradation ($d_{T}$) in Equation (4), which becomes the final degradation feature, is obtained by multiplying the two aspects and avoiding the duplication of the screw speed. The denominators (screw and puller speeds) and numerators (motor load and head pressure) in Equations (2)–(4) increase and decrease, respectively, as the screw wear progresses. To be precise, the head pressure can be maintained during the actual operating process but is not expected to increase.

Consequently, all three aspects of the degradation feature in Equations (2)–(4) are expected to decrease, provided the screw functions properly, even with gradual degradation, as shown in Figure 10. In the figure, the black, blue, and red markers indicate wear levels 1, 2, and 3, respectively. Note that the x-axis ranges from zero to five years. For these plots, the average life span of the screws was assumed to be five years. The time index on the x-axis can vary because the three screws may have different useful lives. The gray curves in Figure 10 depict the expected behavior based on the analysis thus far. The gray star marks the EOL of the screws because screws should be replaced before they deteriorate the catheter quality.

One screw for each wear level was insufficient for validating the degradation feature. Thus, real operational data were applied to the same feature extraction methods, as detailed in the following section.

### 3.3. Real Operational Data Analysis

The corrected degradation features of the screw wear using Equations (2)–(4) are shown in Figure 11. In the figure, each marker represents the mean of the real operational data for a day, and the linear fitting results are shown with black lines. The barrier and Maddock screws represent the control and test groups, respectively. In addition, the cumulative usage time of the system is used for both the barrier and Maddock screws to compare the changes in the monitoring signals according to the degree of screw wear rather than to consider the usage time of screws. The slope values in Figure 11 show that the amount of variation in the degradation features of the Maddock screw is more than twice that of the barrier screw. This is because the cumulative usage time of the barrier screw corresponds to wear level 2 in Figure 10, which shows little degradation of the screw. In Figure 11a, the positive slope indicates that motor degradation is more dominant than screw wear. However, the negative slope of the Maddock screw in Figure 11b clearly shows the screw wear.

However, the distribution of the degradation features fluctuated and was scattered, which was not sufficient to predict the degradation behavior and RUL. Therefore, a weighted cumulative average (WCA) was applied to the degradation features shown in Figure 11b to further highlight the degradation characteristics. The weight in the WCA was defined as the value uniformly divided between 1 and 1/n, where n is the number of data obtained up to the current time. For example, when the current time was 3000 h, n=26 (refer to Figure 11b), and the weights decreased from 1 to 0.0385 (1/n) with a uniform interval of 0.0385 (1/n). Each value of the degradation features in Figure 11b was then multiplied by the calculated weight. Finally, the mean of the weighted feature cumulated from zero to the current time was obtained as the WCA at 3000 h. This process was repeated each time, and the results are shown in Figure 12. As shown in Figure 12a, the WCA of the degradation feature over the entire range contained unstable data with large fluctuations in the early stages of the lifespan. When the full time was reduced after 1000 h, as shown in Figure 12b, the characteristics of each feature could be observed more clearly. The WCA of $d_{M}$ did not show a monotonic trend, which is a basic condition for degradation behavior. However, the WCA of $d_{E}$ and $d_{T}$ decreased continuously, and either of the two could be considered the final degradation feature for the prediction. However, whereas the WCA of $d_{E}$ decreased almost linearly after 1500 h, the rate of decrease of the WCA of $d_{T}$ increased. The behavior of the WCA of $d_{T}$ was an expected characteristic of the general degradation feature. This demonstrated that the degradation feature proposed in Equation (4) was reasonable for describing the wear behavior of the extruder screw. Consequently, the WCA of $d_{T}$ in Figure 12 was extracted from the raw data in Figure 4a and used as the final degradation feature to predict the RUL of the Maddock screw.

## 4. Prediction Result

The RUL of a component measures the time remaining before repair or replacement. In this study, the component of interest was the Maddock screw, and thus the time label in Figure 12 is important. Therefore, the time label of the WCA of $d_{T}$ in Figure 12a was converted to the time scale of the Maddock screw, as shown in Figure 13a. During 4000 h of system operation, Maddock screws were used for approximately 1200 h (more accurately, 1128 h). In the figure, only the solid black dots are considered for RUL prediction because the circle markers up to 400 h indicate the data points where the degradation feature differed from the expected behavior. The final data are depicted by dotted markers in Figure 13b.

Once the degradation data are obtained, a degradation model is assumed for the RUL prediction. Paris and Erdogan [25] proved that crack growth behaves exponentially and developed the Paris model, which is a physical degradation model that describes crack growth behavior under different loading conditions. Goebel et al. [26] used the exponential function as an empirical model to describe battery degradation behavior. Because physical models such as the Paris model are rare, exponential functions are generally employed as empirical degradation models in most prognostic studies. Therefore, in this study, it was assumed that the degradation behavior (z) is described by the following equation:(5)$$z=a+b\times \exp\left(\frac{t}{1000}\right)$$
where a, b, and t are the model parameters and the time, respectively. The model parameters were estimated using the data obtained up to the current time by minimizing the error between the data and model output, z. There are several methods for parameter estimation, and the Bayesian method [27] was used in this study. In the Bayesian method, the model parameters are estimated in the form of a probability density function (PDF). Subsequently, the Markov chain Monte Carlo sampling method was employed to draw samples from the PDF of the model parameters. Consequently, the final model output at time t was obtained by substituting the estimated values (drawn samples) for a and b in Equation (5). For example, the gray curve in Figure 13b is the model output at EOL obtained using all data and the Bayesian method. Because all monitoring data were used, the curve was considered to be the true degradation behavior, excluding the noise in the data. More details on the Bayesian method and implementation code can be found in the book by Kim et al. [27]. Next, a degradation threshold should be determined by considering the trade-off between risk and cost. However, determining the threshold is another specialized field; thus, in this study, the threshold was assumed to be the true wear level at EOL. In Figure 13b, the green horizontal line represents the threshold of 7.24, where the true model (gray curve) reaches an EOL of 1128 h (red vertical line).

The prognostics can be performed using the above information and assumptions, that is, the data, degradation model, and threshold. The prediction results of the degradation behavior at 768 h are shown in Figure 14a. In the figure, the dotted markers represent the data obtained up to the current 768 h, which were used to estimate the model parameters in Equation (5). The dashed and dotted red curves represent the median and 90% interval of the degradation prediction results, respectively, with the estimated model parameters. The EOL prediction is distributed between 1050 h and 1200 h, which is when the red curves reach the threshold (green horizontal line). The RUL was predicted by subtracting the current time from the predicted EOL, as indicated by the magenta vertical line in Figure 14b. The RUL prediction results were obtained by repeating the process for the degradation prediction in Figure 14a every time data was obtained. The results in Figure 14b show that the RUL prediction results (red lines) are very close to the true results (gray diagonal line) with narrow distributions after approximately 750 h. In other words, an accurate prediction is possible approximately 380 h before the EOL. The extrusion system is operated for approximately eight hours a day on weekdays; thus, the maintenance and/or replacement of Maddock screws can be scheduled approximately two months before EOL (it is a twofold improvement in the prediction of the RUL for the system using both barrier and Maddock screws).

## 5. Conclusions

In this study, a novel method for degradation feature extraction is proposed to predict the RUL of an extrusion screw using real operational data. The micro-extrusion system has been operational for more than five years, and in the last two years real operational data were obtained while the system produced various specifications of medical catheters. During this period, the Maddock screw reached its EOL owing to severe wear and was targeted for RUL prediction based on the proposed degradation feature. The proposed degradation feature exhibited typical characteristics of degradation behavior, which monotonically decreased with an increase in the degradation rate over time. The degradation feature was used to predict the RUL of the Maddock screw, and accurate results for the RUL were obtained approximately 380 h before EOL.

The main objective of this study was to accomplish the prognostics process using real operational data from a micro-extrusion system in service. First, real operational data are invaluable compared with test-level data, which are usually obtained at the component level under severe load conditions and well-planned operation conditions. Next, a valid degradation feature was extracted from the system-level data based on the fusion of the physical interpretation by the authors and the data information. Notably, the target object for prognostics was at the component level; however, the data used in this study were obtained at the system level, which reflects multiple complex aspects of the system within one type of signal. Finally, a prognostic study for the extrusion system was addressed, not by common objects such as bearings and batteries.

This study had some limitations. First, the proposed method for degradation feature extraction has not been fully validated owing to a lack of operational data. However, real operational data are still being collected and will be used in further studies. Next, the degradation of other components, including the motor, was ignored because screw wear is currently predominant. The degradation features of other components will become distinct over time, and they will be applied to improve the degradation feature of the screw. Lastly, the prognostics study for other components will be conducted aiming for system-level prognostics.

## Figures and Tables

**Figure 1 sensors-23-00637-f001:**
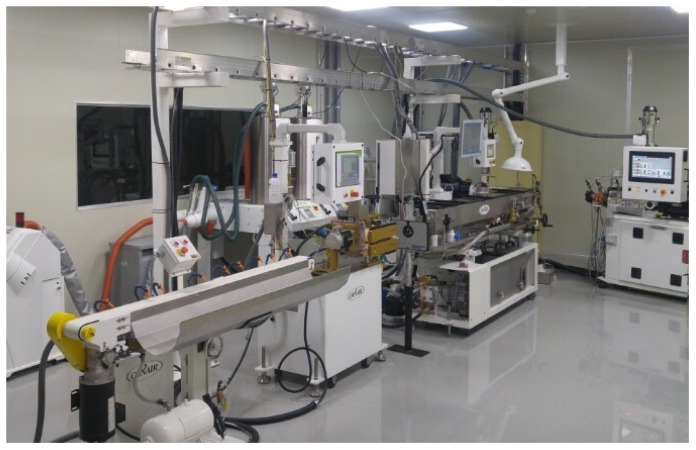
Medical multi-lumen tubing extrusion system.

**Figure 2 sensors-23-00637-f002:**
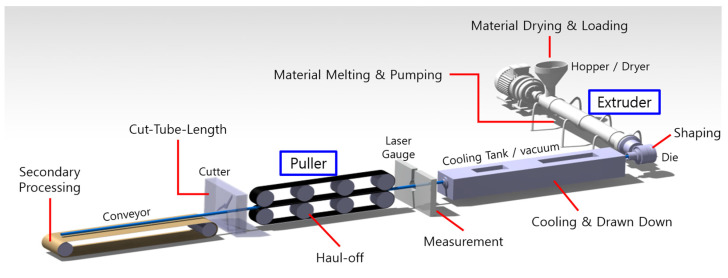
Illustration of the entire extrusion system.

**Figure 3 sensors-23-00637-f003:**
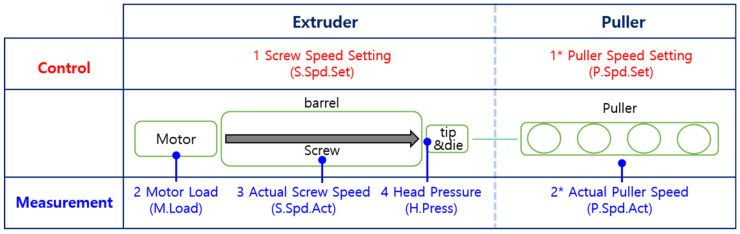
Screw wear-related parts and sensing location.

**Figure 4 sensors-23-00637-f004:**
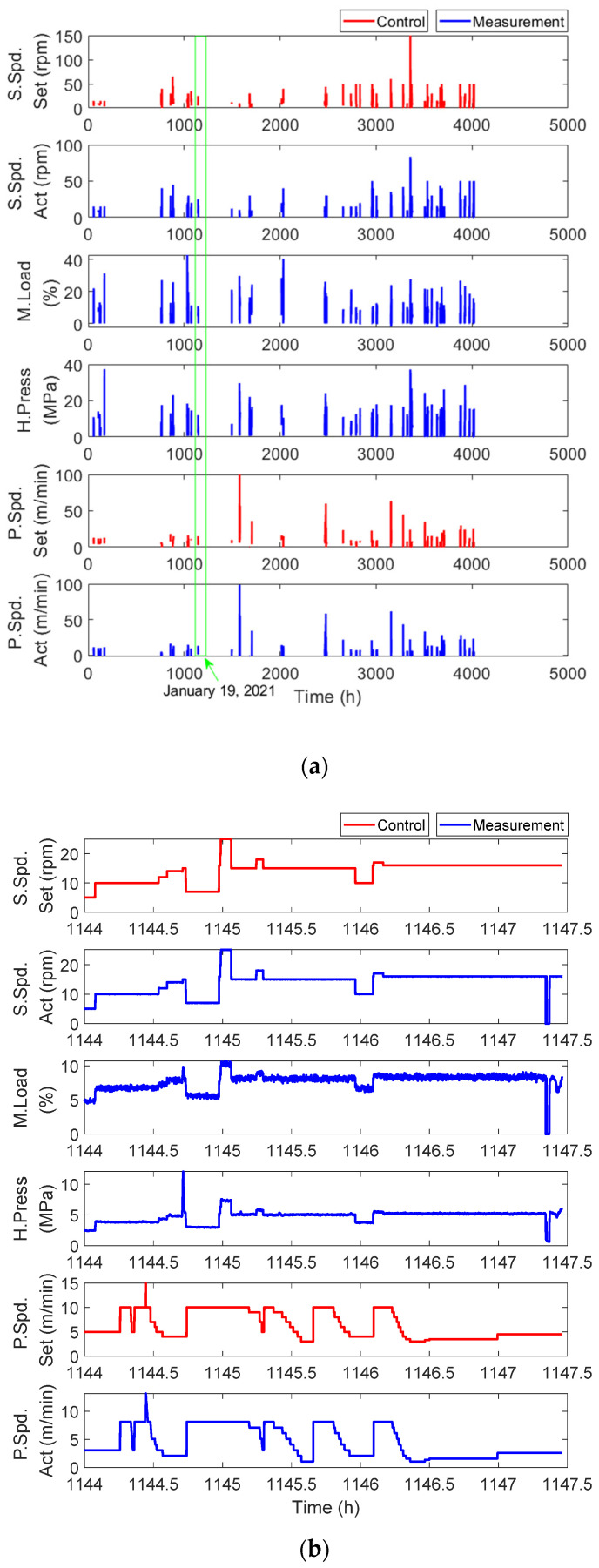
Real operational data: (**a**) All measurements for the Maddock screw; (**b**) Extrusion data for the Maddock screw collected on 19 January 2021.

**Figure 5 sensors-23-00637-f005:**
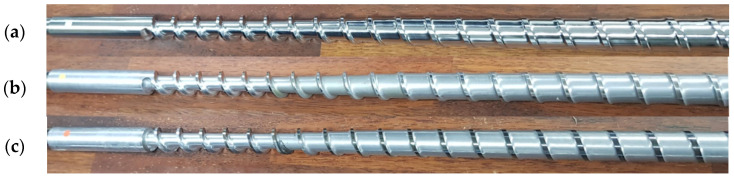
Three screws used in the screw wear experiments: (**a**) wear level 1: intact; (**b**) wear level 2: moderate wear; and (**c**) wear level 3: severe wear.

**Figure 6 sensors-23-00637-f006:**
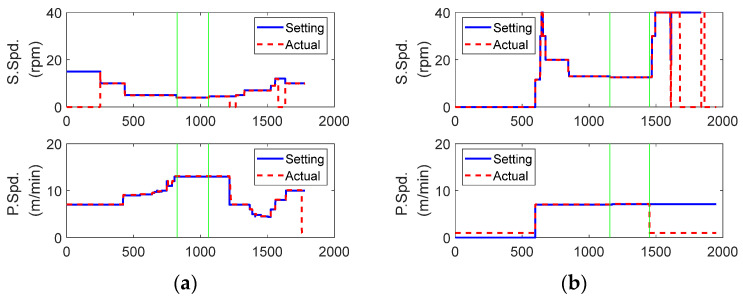
Difference between setting and actual speeds (Maddock): (**a**) data monitored on the first day (6 July 2020); (**b**) data monitored on the last day (25 March 2022).

**Figure 7 sensors-23-00637-f007:**
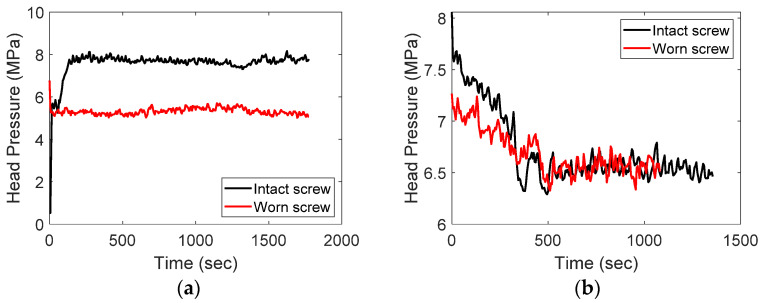
Head pressure change according to screw wear: (**a**) no catheter production; (**b**) during catheter production.

**Figure 8 sensors-23-00637-f008:**
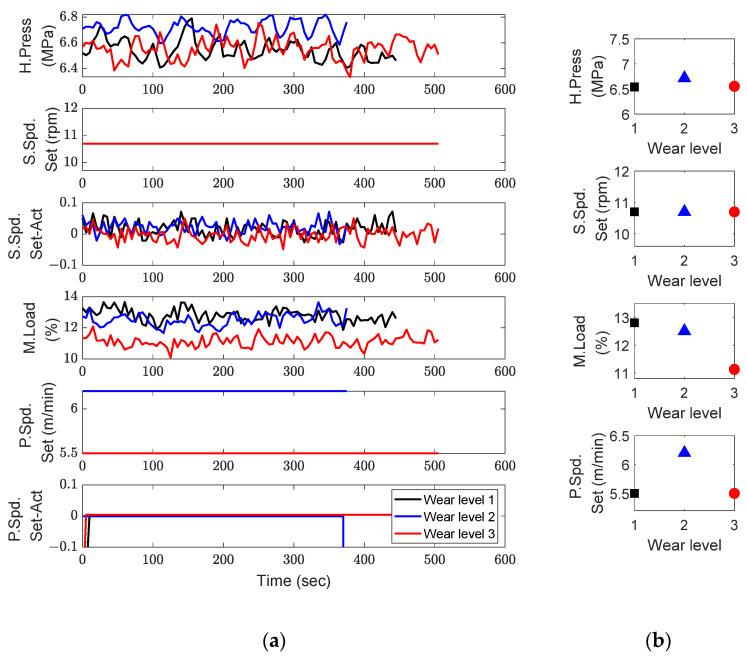
Experimental data according to the level of screw wear, keeping catheter production and materials the same: (**a**) operational data; (**b**) mean of the operational data.

**Figure 9 sensors-23-00637-f009:**
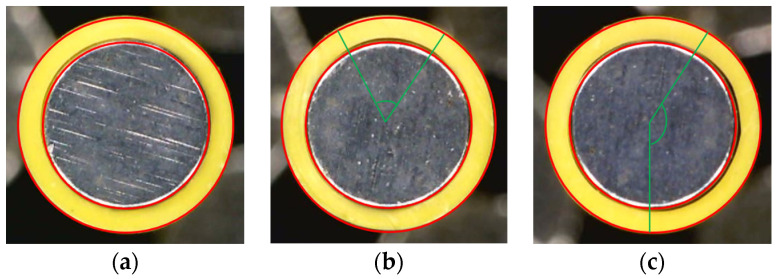
Quality of catheters according to the level of screw wear: (**a**) wear level 1; (**b**) wear level 2; (**c**) wear level 3.

**Figure 10 sensors-23-00637-f010:**
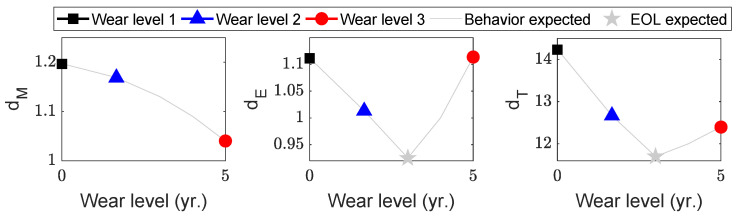
Degradation feature analysis; the material and product were kept the same.

**Figure 11 sensors-23-00637-f011:**
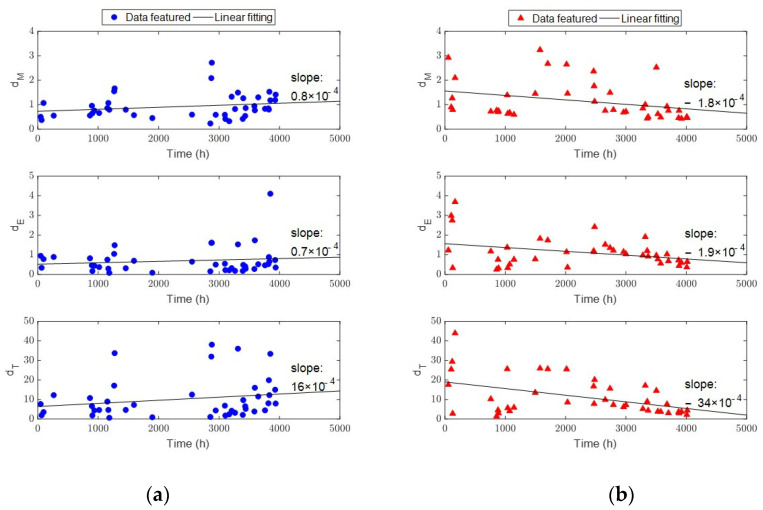
Degradation feature with correction factor using Equations (2)–(4): (**a**) barrier screw; (**b**) Maddock screw.

**Figure 12 sensors-23-00637-f012:**
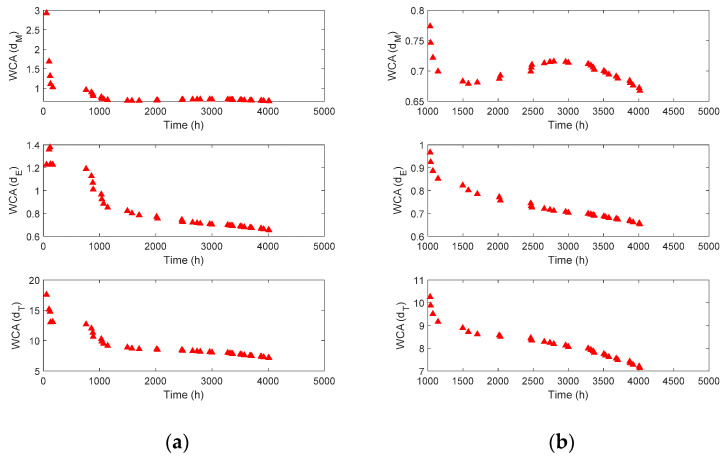
Degradation feature analysis based on proposed methods: (**a**) entire time scale; (**b**) after 1000 h.

**Figure 13 sensors-23-00637-f013:**
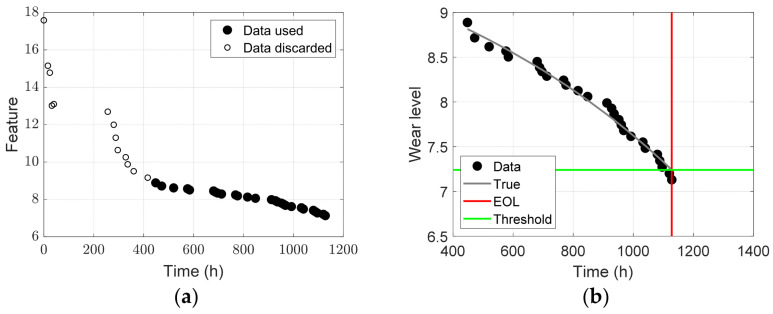
Given information and assumption for prognostics: (**a**) data for RUL prediction; (**b**) assumed true model and threshold.

**Figure 14 sensors-23-00637-f014:**
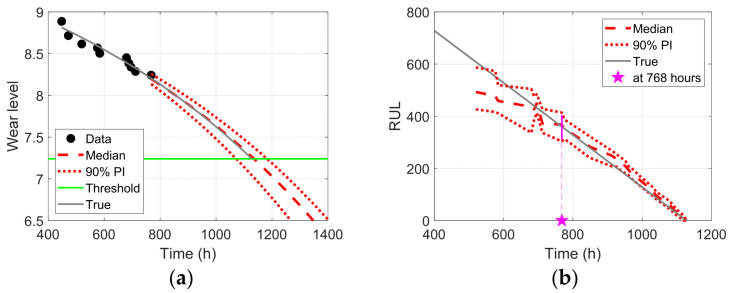
Prognostics results for the Maddock screw: (**a**) degradation prediction at 768 h; (**b**) RUL prediction for entire time.

**Table 1 sensors-23-00637-t001:** Results of the flow rate measurement test according to screw wear level.

						Unit: [g/min]	
Screw Wear Level	Test 1	Test 2	Test 3	Average		Standard Deviation	
				Amount	Ratio to Wear L. 1	Amount	Ratio to Wear L. 1
1	13.88	13.85	13.89	13.87	-	0.021	-
2	15.67	15.64	15.66	15.66	12.9%	0.015	−28.6%
3	14.17	14.27	14.48	14.30	3.1%	0.158	652%

## Data Availability

Not applicable.

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
