# Peer review of "Degradation Feature Extraction Method for Prognostics of an Extruder Screw Using Multi-Source Monitoring Data"

_sensors, 2023, doi:10.3390/s23020637_

Round 1

Reviewer 1 Report

(1)     Generally speaking, this paper lacks some theoretical and methodological innovations. the proposed method only can give some empirical features. And the proposed method may only be applicable to the specific micro-extrusion system. The generalizability of the method proposed in this paper is my major concern.

(2)     The English of this paper has affected the reading and understanding of the paper, and the structure of the entire paper should be greatly improved or even rewritten.

(3)     The introduction of the proposed method in the abstract is too brief and general, and does not reflect the innovation of the proposed method, which must be improved.

(4)     It is mentioned several times in the paper that analyzing real operating data is more challenging than analyzing laboratory data. However, this paper lacks a specific description of these challenges, such as the greater uncertainty of the actual environmental conditions, the more complex degradation modes, or the mutual interference of multiple components. At the same time, this paper does not theoretically explain how the proposed method can effectively solve these challenges.

Author Response

The authors greatly appreciate the reviewer’s valuable time in reviewing the manuscript. Please check the file attached for the authors’ response to the reviewer’s comments.

Reviewer 2 Report

1. FIG. 1 and FIG. 2 are not clear with low resolution;

2. How to determine the threshold used in the prediction of RUL?

3. Units need to be added to the vertical coordinates in FiG.4 and FiG.6.

4. FIG. 9 does not show the difference of three wear levels.

Author Response

(The authors gave the same response as above.)

Reviewer 3 Report

The paper has two objectives. First, the aim is to extract the degradation feature to monitor the wear of the screw used in a real extrusion system and, second, the extracted feature is used to predict the RUL of the screw.

The study should be presented in a more clear way. As far as I understand, the authors consider two different datasets: experimental data obtained under controlled conditions in a laboratory (I will refer to this dataset as dataset 1), and real operational data (which will be referred as dataset 2).

The authors use dataset 1 to extract the degradations feature based on the  measurements reported by a set of sensors that monitor some components of the system and provide useful information from which the following is selected: motor load, head pressure, and screw and puller speeds.

From the analysis of dataset 1 the authors propose a function that properly measures the screw wearout. Then, this function is considered using dataset 2 to obtain a measure of the screw degradation in real operational conditions.

This is the first objective of the authors and it covers most of the manuscript (12 pages out of 15). I will not discuss much of this part of the paper since I am not familiar with this kind of industrial processes. However I will ask some questions later that I would like to be clarified.

I will focus next on the second objective: to predict the RUL based on degradation. In this respect I have the following concerns:

1.    The authors assume an exponential relationship between degradation an time, according to Eq (5). This relationship is not well justified. There are no further explanation in the paper nor any references to support it. The authors must justify this point properly. I would recommend the authors to look at literature on accelerated life tests.

2.    The authors mention that a Bayesian method is used to fit this model. More details are required here also. Please explain the method. What do the authors mean with the sentence ‘The curve represents the true degradation behavior, whereas the data also contains noise’

3.    A threshold level is considered for degradation to estimate the EOL. This threshold is taken as the true wear value at the EOL. This value is chosen ad hoc, a more general discussion is necessary here.

 My conclusion is that this part of the paper must be worked harder.

General comments:

1.       All conclusions in the paper are based on descriptive analysis of the data. For example trends of degradation features in Figure 11 are validated by just looking at the data. A more deep statistical analysis should be carried out to infer an adequate model for the data.

2.        Please explain the weighted cumulative average applied to generate Figure 12.

3.       Figure 4 is not ease to interpret. To better understand the data, this should be accompanied by a proper statistical summary.

4.  Eq. (1) gives the screw wear as a decreasing function of time. The notation here is confusing. It is not ‘understandable’ that wear decreases with time. This point should be considered here and also in Eq. (2)-(4).

5.  Why the duplication of screw speed is not considered in Eq (4)? Please justify.

6. No references are quoted in Section 3 to support the methodology. Please provide, or  properly justify the arguments presented.

7. The flow rate test should be specified (page 9). What is ‘g’ in Table 1?

Author Response

(The authors gave the same response as above.)

Round 2

Reviewer 3 Report

The authors have responded to all concerns in my previous report and I consider the manuscript has been sufficiently improved.